# Internal Nasal Valve Modification via Correction of High Dorsal Deviation Using a Modified Mattress Suture Technique

**DOI:** 10.3390/jcm11195888

**Published:** 2022-10-05

**Authors:** Joongbo Shin, Jungkyu Cho, Sang Duk Hong, Yong Gi Jung, Gwanghui Ryu, Hyo Yeol Kim

**Affiliations:** Department of Otorhinolaryngology-Head and Neck Surgery, Sungkyunkwan University School of Medicine, Samsung Medical Center, Seoul 06351, Korea

**Keywords:** septoplasty, rhinoplasty, suture techniques, internal nasal valve

## Abstract

High dorsal deviation of the septum can cause nasal obstruction due to internal nasal valve (INV) stenosis. We have developed a new technique using a modified mattress suture on the bony-cartilaginous junction to correct high dorsal septal deviation. This study focused on the effect of this suturing technique on the modification of the INV. We enrolled 40 patients who underwent septoplasty using a modified mattress suture technique. We retrospectively analyzed the data of the preoperative and postoperative INV angles and cross-sectional areas (CSAs), which were measured using computed tomography. In addition, we compared the patients’ subjective nasal symptoms, which were measured with the preoperative and postoperative Nasal Obstruction Symptom Evaluation (NOSE) instrument. Postoperative increases in the narrow side INV angle and CSA were achieved. Additionally, the wide side INV angle and CSA were significantly decreased postoperatively. The INV and CSA ratio (wide/narrow) were also decreased postoperatively and were brought closer to 1. The subjective nasal symptoms also exhibited significantly reduced NOSE values. In this study, we confirmed the effects of septoplasty using a modified mattress suture technique for INV modification through the comparison of the preoperative and postoperative INV angles and CSAs.

## 1. Introduction

For diagnosis and clarifying anatomical terminology, Cottle (1961) proposed to divide the internal nose into five areas: area 1, nostril; area 2, nasal valve; area 3, area underneath the bony and cartilaginous vault; area 4, anterior part of the nasal cavity; area 5, posterior part of the nasal cavity [1]. The internal nasal valve (INV) is the narrowest part of the nasal airway, and its borders are the caudal margin of the upper lateral cartilage, the nasal septum, and the anterior border of the inferior turbinate [2]. The normal angle of the INV is 10 to 15 degrees in Caucasians [3]. Since the INV is the narrowest region of the nose, this area has the greatest influence on airflow resistance [4]. Nasal obstruction due to nasal valve collapse can be static, dynamic, or both [5]. Dynamic collapse can occur due to Bernoulli’s principle when the lateral nasal wall has insufficient support [3]. Stenotic narrowing or stenosis of the INV is caused by anatomical problems, such as septal deviation or turbinate hypertrophy [5]. The nasal septum is an important unit of nasal function, and it plays an important role in maintaining respiratory and nasal physiology. Septal deviation of the nose is one of the most common disorders encountered in otorhinolaryngology and one of the major causes of nasal obstruction [6]. The locations of septal deviation are distributed into five parts: caudal, dorsal, midseptal, bony septum, and inferior septum [7]. Among these, a dorsal septal strut is consistent with the perpendicular plate of the ethmoid bone and the upper dorsal portion of the cartilaginous septum; this area is a part of the INV. However, a dorsal deviation is the most difficult and delicate issue to correct surgically. A previous study showed that deflection of the dorsal septum was observed in 65% patients with INV narrowing [8]. In addition, one of the most common areas where deviation persists after septoplasty is the dorsal septum [7]. Therefore, an adequate correction of dorsal septum deflection is essential for improving both symptoms and patient satisfaction in patients with INV narrowing. We previously reported that a “modified mattress suture technique” was a useful technique for correction of high dorsal deviation of the septum around the INV [9]. The minimal cross-sectional area (CSA) of the INV was also significantly improved. In the present study, we aimed to explore the effect of a modified mattress suturing technique to correct high dorsal septal deviation on the INV. We measured the INV angle and CSA of both the narrow and wide sides using preoperative and postoperative computed tomography (CT) in patients with high dorsal deviation.

## 2. Materials and Methods

### 2.1. Patients

We retrospectively reviewed the medical records of patients who had nasal valve stenosis due to deviation of the dorsal septum and underwent correction of high dorsal deviation of the nasal septum using a modified mattress suture from November 2017 to December 2019. We diagnosed high dorsal deviation by endoscopic nasal examination and CT evaluation. Eligible participants were screened from the database of a single surgeon’s (H.Y.K.) rhinology practice at a tertiary referral hospital.

We excluded people who underwent revision septoplasty (*n* = 10), endoscopic sinus surgery (*n* = 8), or rhinoplasty (*n* = 5) at the same time. We collected the demographic data (including age, sex, past medical history, and previous surgery) of the remaining participants. The patients underwent both endoscopic examination and acoustic rhinometry (AR) testing preoperatively. Their preoperative and postoperative CT scans were also evaluated. Subjective symptoms were evaluated using the Nasal Obstruction Symptom Evaluation (NOSE) scale. All patients underwent bilateral turbinoplasty at the same time.

This study was approved by the Institutional Review Board (No. 2022-05-098) and the requirement for informed consent was waived.

### 2.2. Measurement of the INV Angle and CSA

To objectively assess the correction of the high dorsal septal deviation, CT scanning was performed both before the surgery and after the surgery. We measured the INV angle and CSA as previously described [10,11,12]. For each individual, the distance between the nostril and INV as determined by AR was marked on a sagittal CT image. Afterwards, we obtained a coronal section perpendicular to the acoustic axis passing through the aforementioned point (Figure 1). The INV angle was measured, and then the irregularities in the medial and lateral walls were averaged (Figure 2A); the CSA was measured using the outer margin of the air passage (Figure 2B) in this representative section using an Amira (Mercury Computer Systems/3D Viz group, San Diego, CA, USA).

### 2.3. Surgical Procedure

The detailed surgical technique was described in a previous article [9]. After elevating the mucoperichondrial flap from the concave side of the nasal septum, the middle portions of the deflected bony septum and any surplus cartilage were removed. Partial thickness scoring was performed on the concave side of the cartilaginous septum. Then, a drill was used on the upper part of the bony septum to create two suture holes, and a greenstick fracture was created. A modified mattress suture was performed on the upper part of the cartilaginous septum and the two holes to correct the deflected nasal septum in the midline (Figure 3). If the deviation of PPE is present, it is difficult to correct the cartilaginous septum connected to the PPE. Therefore, the purpose of this procedure is to correct the PPE through a greenstick fracture and to straighten, fix, and stabilize the corrected PPE and the deviated upper part of the cartilaginous septum and the whole dorsal septum with thread tension.

### 2.4. Statistical Analysis

The statistical analysis was performed using the Statistical Package for the Social Sciences (SPSS) software (Version 28, SPSS, Inc., Chicago, IL, USA); values of *p* < 0.05 were considered statistically significant. The differences in the preoperative and postoperative INV angles and the CSAs were analyzed using the Wilcoxon signed-rank test. The ratios of the wide INV angle and the narrow INV angle and the wide CSA and the narrow CSA were also analyzed using the same method.

## 3. Results

Forty patients were included in this study (Table 1). In total, 34 patients (85%) were male, and 6 (15%) were female. The mean age was 38.1 years (range: 18–70 years). The preoperative nasal septal examination demonstrated mild septal deviation in 12 patients (30%), moderate septal deviation in 26 patients (65%), and severe septal deviation in 2 patients (5%). The postoperative evaluation (CT, NOSE scale) interval ranged from 2.43–3.77 months (mean: 3.23 months).

We compared the preoperative and postoperative means of the INV angle and CSA using CT scanning (Table 2). The narrow INV was widened (from 11.80 ± 2.43 degrees before the surgery to 13.33 ± 2.35 degrees after the surgery, *p* = 0.002), the wide INV was narrowed (from 19.89 ± 4.80 degrees to 17.49 ± 3.04 degrees, *p* = 0.002), and the difference between narrow and wide decreased (from 8.09 ± 4.43 degrees to 4.60 ± 3.02 degrees, *p* < 0.001) postoperatively. Similarly, the narrow CSA was widened (from 0.66 ± 0.18 cm^2^ before the surgery to 0.78 ± 0.27 cm^2^ after the surgery, *p* = 0.011), the wide CSA was narrowed (from 1.30 ± 0.28 cm^2^ to 1.20 ± 0.31 cm^2^, *p* = 0.001), and the difference between narrow and wide decreased (from 0.64 ± 0.28 cm^2^ to 0.47 ± 0.30 cm^2^, respectively, *p* = 0.002) postoperatively.

After comparing the ratios of the wide parts to the narrow parts of the INV angles and CSAs, we determined that the ratios were closer to 1 after the surgery; the ratio of the INV angle between the wide and narrow sides changed from 1.73 ± 0.52 before the surgery to 1.35 ± 0.31 after the surgery (*p* < 0.001), respectively, while that of the CSA changed from 2.12 ± 0.89 to 1.72 ± 0.89 (*p* = 0.004).

The patients’ subjective symptoms also showed significantly reduced NOSE values from 58.13 before surgery to 25.63 after surgery (*p* < 0.001; Table 3). All sub-items of the NOSE questionnaire also showed a significant decrease postoperatively.

The preoperative endoscopic examination shows septal deviation and asymmetry of the bilateral nasal cavity (Figure 4A). After surgery, symmetric changes in the bilateral nasal cavities were observed (Figure 4B). In comparison of the preoperative and postoperative CT images, the deviated PPE was corrected (Figure 4C) and the deviated upper part of cartilaginous septum involved in the INV was corrected (Figure 4D) after surgery.

## 4. Discussion

Many studies on nasal valve physiology and structure have been conducted in the past [13,14]. Small anatomic disturbances in the nasal valve can produce significant functional problems by narrowing the nasal valve angle [15]. To solve this problem, various treatments have been proposed through several studies. Most of the previous treatments corrected the nasal valve by widening the space between the upper lateral cartilage and the septum, such as a flaring suture or a spreader graft through dorsal approach [16,17,18].

In this study, we demonstrated that the INV compromise was successfully corrected using a modified mattress suture technique of the bony-cartilaginous junction. In addition, the patients’ subjective nasal symptoms were relieved after surgery. Although procedures accompanied by septoplasty (such as turbinoplasty) may affect the CSA, this effect can be excluded because all accompanying procedures were performed equally in enrolled patients. Kang et al. showed the effect of a modified mattress suture by comparing the CSA measured using acoustic rhinometry [9]. However, we compared both the CSAs and INV angles in the INV area using an Amira and demonstrated the effect of a modified mattress suture.

When evaluating septal deviation, high dorsal deviation of the nasal septum around the bony-cartilaginous junction (“keystone area”) is often overlooked by focusing only on caudal septal deviation or bony septal deviation because it is difficult to approach and correct this area using previous endoscopic approaches [9]. Traditionally, widening of the INV can be accomplished using spreader grafts, spreader flaps, butterfly grafts, or batten grafts, which are typically performed via external nasal approaches [19]. This modified mattress suture technique of the bony-cartilaginous junction practices a greenstick fracture to the deviated perpendicular plate of the ethmoid bone and cartilage and then pulls and unfolds them with a horizontal mattress suture. Therefore, it is a useful and simple method that can be easily handled via an endoscopic approach and can rapidly fix a high dorsal septal deviation around the keystone area. With this surgical technique, we achieved the desired INV angle and CSA symmetry by the correction of high dorsal deviation.

The improvement in patients’ subjective nasal symptoms was evaluated using the NOSE scale. There was a significant improvement in the clinical parameters of nasal obstruction after the surgery. In addition, we could also objectively confirm the technique’s success using the preoperative and postoperative CT scans, which showed that the INV angles and the CSAs of the narrow sides were widened after the surgery and that the wide sides had been narrowed. Furthermore, the ratios of the narrow to the wide portions of the INV angles and CSAs moved closer to 1. Based on these results, we confirmed that the objective as well as the subjective findings improved significantly after surgery.

This study had some limitations. It included a relatively small number of participants and had no control group because it was difficult to set up a control group to compare the different surgical methods. This study also was not able to consider the effect of mucosal factors and the nasal cycle during CT scanning, which may have affected results. In addition, the results of this study were limited to improvements in internal valve stenosis and did not produce improvements in dynamic compromise. This finding could not be hypothesized to any races other than an Asian population. This finding requires further verification in other races because there are racial differences in the characteristics of the INV structure.

Despite these limitations, this study showed that the modified mattress suture technique of the bony-cartilaginous junction is a useful and simple technique, and anyone can easily use this surgical technique to correct high dorsal septal deviation and INV stenosis.

## 5. Conclusions

This study showed that a modified mattress suture technique of the bony-cartilaginous junction could be used to correct easily high dorsal deviation around the keystone area via an endonasal approach, which cannot be corrected using previous septoplasty techniques. We also confirmed the effects of the INV modification and improvement in patients’ subjective symptoms but did not prove the relationships between the improvement of the INV angle, the CSA, and the subjective symptoms. In the future, we will conduct further studies including additional tests such as PNIF to identify this relationship.

## Figures and Tables

**Figure 1 jcm-11-05888-f001:**
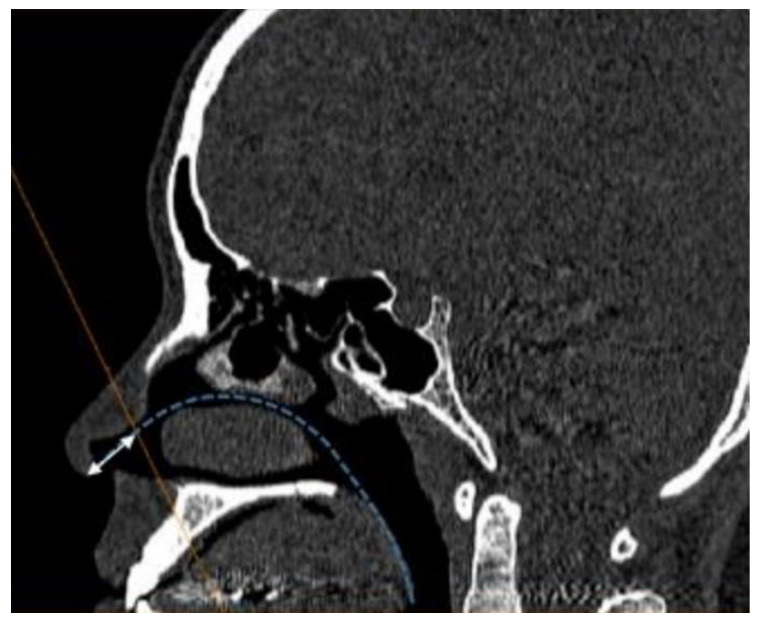
Identification of the plane perpendicular to the imaginary acoustic axis. Dotted blue line: the imaginary acoustic axis; white double-headed arrow: the distance from the nostril to the internal nasal valve (INV) obtained through AR; orange line: the plane perpendicular to the imaginary acoustic axis.

**Figure 2 jcm-11-05888-f002:**
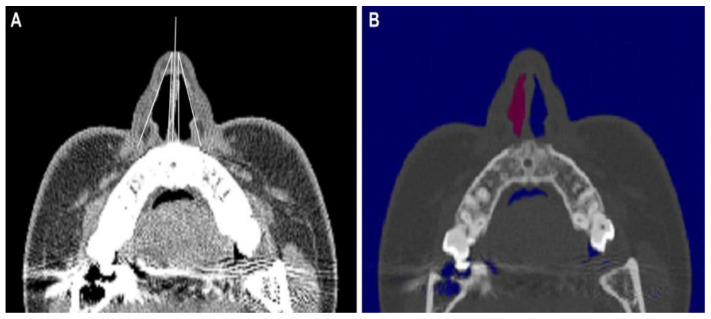
(**A**) Measurements of the INV obtained via computed tomography (CT). The angle of the INV was measured by averaging the irregularities in the medial and lateral walls. (**B**) Measurements of the cross-sectional area (CSA) via CT. The nasal valve area was measured in images taken perpendicular to the acoustic axis.

**Figure 3 jcm-11-05888-f003:**
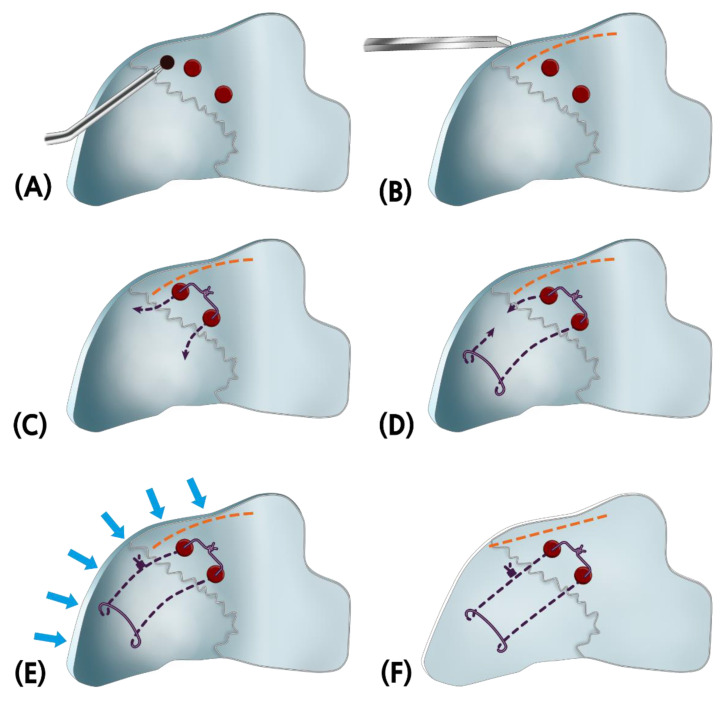
Illustration of a modified mattress suture technique of the bony-cartilaginous junction below the keystone area. (**A**) Vertical holes are made into the perpendicular plate of the ethmoid bone (PPE) using a 2.5-mm drill. (**B**) Using a 2-mm osteotome, a greenstick fracture is performed on the dorsal part rather than the hole in the PPE. (**C**) Vicryl suture with double-arm needles is passed through the two holes, from the concave to the convex sides of the nasal cavity. (**D**) The lower needle is brought down, parallel to the nasal dorsum, and then passed through the septal cartilage. (**E**) From the convex side of the nasal cavity, the suture is tied up while pushing the cartilage and the bone. (**F**) The nasal septum is now straightened, starting below fracture line.

**Figure 4 jcm-11-05888-f004:**
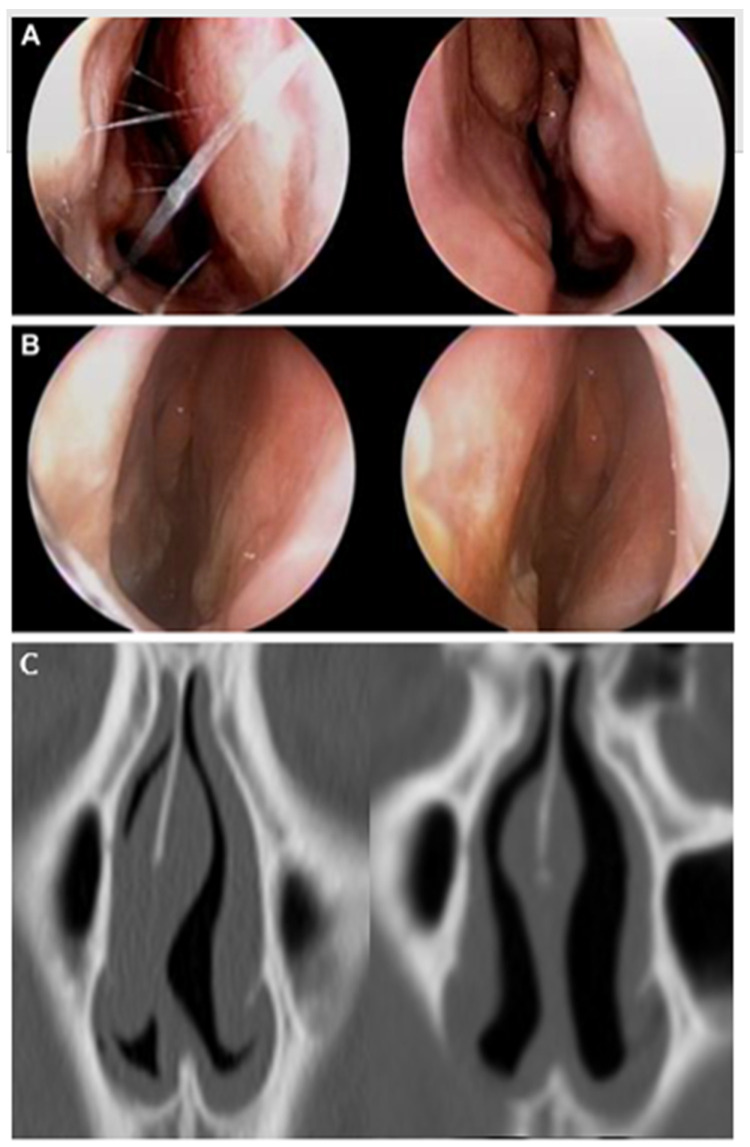
Representative endoscopic examination and CT imaging. (**A**) A preoperative endoscopic examination. (**B**) The postoperative endoscopic examination shows symmetric changes in bilateral nasal cavities. (**C**,**D**) Preoperative and postoperative CT images show corrected the PPE and upper part of cartilaginous septum after surgery.

**Table 1 jcm-11-05888-t001:** Demographic and clinical data.

Characteristic	Value
Sample size	40
Age, mean (range)	38.1 (18–70)
Sex, n (%)	
Female	6 (15)
Male	34 (85)
History of nasal trauma, n (%)	8 (20)
Deviation side, n (%)	
Right	25 (62.5)
Left	15 (37.5)
Types of severity for septal deviation, n (%)	
Mild	12 (30)
Moderate	26 (65)
Severe	2 (5)
Postoperative evaluation interval–CT, NOSE scale (range, month)	3.2 (2.4–3.8)

**Table 2 jcm-11-05888-t002:** Comparison of the preoperative and postoperative mean of the INV angle and CSA.

	Preoperative, Mean (SD)	Postoperative, Mean (SD)	*p*-Value
INV angle (degree)			
Narrow angle	11.80 (2.43)	13.33 (2.35)	0.002
Wide angle	19.89 (4.80)	17.49 (3.04)	0.002
Difference (wide–narrow)	8.09 (4.43)	4.60 (3.02)	<0.001
Ratio (wide/narrow)	1.73 (0.52)	1.35 (0.31)	<0.001
CSA (cm^2^)			
Narrow CSA	0.66 (0.18)	0.78 (0.27)	0.011
Wide CSA	1.30 (0.28)	1.20 (0.31)	0.001
Difference (wide–narrow)	0.64 (0.28)	0.47 (0.30)	0.002
Ratio (wide/narrow)	2.12 (0.89)	1.72 (0.89)	0.004

CSA, cross-sectional area; INV, internal nasal valve; SD, standard deviation.

**Table 3 jcm-11-05888-t003:** Symptom improvement assessed by the Nasal Obstruction Symptom Evaluation (NOSE) questionnaire.

	Preoperative, Mean (SD)	Postoperative, Mean (SD)	*p*-Value
NOSE, total	58.13 (10.48)	25.63 (10.33)	<0.001
Nasal congestion or stuffiness	16.63 (3.65)	7.13 (3.74)	<0.001
Nasal blockage or obstruction	16.50 (3.62)	6.38 (4.38)	<0.001
Trouble breathing through my nose	10.13 (5.49)	4.88 (4.16)	0.002
Trouble sleeping	7.13 (5.30)	3.75 (3.15)	0.002
Unable to get enough air through my nose during exercise or exertion	7.75 (5.88)	3.50 (4.11)	<0.001

## Data Availability

The data presented in this study are available on request from the corresponding author. The data are not publicly available due to ethical concerns.

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
