# Peer review of "Internal Nasal Valve Modification via Correction of High Dorsal Deviation Using a Modified Mattress Suture Technique"

_jcm, 2022, doi:10.3390/jcm11195888_

Round 1

Reviewer 1 Report

1 - General comment: the authors should avoid self citation "we", "our" in their paper

2 - Biblio references: we are deeply sorry not to find any reference to the basic fundamental works done on the nasal valve in the history (before 2000), as this is a highly sensible topic in rhinology.

3 - In the abstract and in the text: high septal deviation is not responsible of INV collapse but may provoke INV stenosis, that is a different mechanism. 

4 - Introduction:

Definition of septal deviation is mandatory. Are we talking about axis deviation, C-shape deviation or frontal deviation?

Classification of septum areas (or segments) has been done by Cottle. No reference is found about this classification. Line 40: the risk of "saddle nose" after K-Area manipulation is not appropriate in this article that focuses on deviations. Otherwise this could be a subject of discussion

5 - In the text (method),

Surgical: a better summary or a drawing of the technique proposed by the authors is highly recommended as the reader is not supposed having read the previous paper written by the authors. The information given don't allow for understanding the surgical procedure. Informations are required for the interpretation of the motivation for and the role of the "greenstick fracture" and the holes drilled in the bones in the principles of the technique as this is anatomically far from the INV but may be useful for stabilization of the dorsal septum. This should have also allowed for better interpretation of the results presented here. Moreover, what are the other surgical procedures done on the rest of the septum? Fracturing the PPE and repositioning it during septoplasty is a very well known procedure already presented in old papers and textbooks. Fixation is a new way. However, how are the authors controlling intraoperatively the exact position (vertical) of the dorsal septum, as the bleeding in the operative field may disturb clinical and endoscopic evaluations?

Turbinoplasty: the authors state that the same procedures have been done on both sides. However, the role of turbinoplasty in the functional evaluation is critical. A group of patients getting turbinoplasty alone (following the same turbinoplasty techniques that should have been described by the way, as there are many procedures that can be done on the inferior turbinates) should have been critical in the functional evaluation. Please comment.

6 - Fig3 Results: "C" shows a CT cross-section at the level of the PPE and nasal bones that is never involved in the INV. The PPE fracture is well shown. Is this the "greenstick fracture" introduced in the surgical method? Moreover, these 2 CT sections are not done at the same level as observed on the inferior turbinate.

Limitations: The functional evaluation is global (both right and left sides) and there is no possibility to differentiate the results related to septoplasty or turbinoplasty or even the narrow side vs the wide side. Please comment.

7 - In the conclusion, the statement that the technique improves the breathing function is not supported by objective datas. The improvement of the CSA is greatly interesting and nicely shown but doesn't allow to make the conclusion of the efficacy on the nasal breathing function. Please comment the relationships between CSA,INV and functional evaluation.

However, the CT measurements analysing the high dorsal septum are interesting to show the efficacy of the surgical procedure on the anatomy of the dorsal septum. This is the only conclusion that can be presented here due to the lack of functional proofs related to the single septoplasty procedure.

Reviewer 2 Report

This is a well-written manuscript. Several comments are suggested.

1. One illustration of the surgical technique may help for understanding.

2. When did the patients have the post-OP evaluation (CT scan, NOSE scale)?

3. How to define the severity of septal deviation (mild, moderate, severe) in this study?

4. Do the authors have the data of long-term follow-up results of the patients receiving this surgical method for correcting high dorsal septal deviation?
